# Prognostic nomogram integrated with inflammatory marker ratios for assessing in-hospital mortality risk in patients with acute type A aortic dissection

Xiao-Chai Lv[1,2,3☉], Lin-feng Xie[1,2,3☉], Min-xia Xie[1,2,3], Lei Wang[1,2,3], Yan-ting Hou[1,2,3], Liang-wan Chen [1,2,3]*

1 Department of Cardiovascular Surgery, Fujian Medical University Union Hospital, Fuzhou, Fujian, PR China, 2 Key Laboratory of Cardio-Thoracic Surgery (Fujian Medical University), Fujian Province University, Fuzhou, Fujian, PR China, 3 Fujian Provincial Center for Cardiovascular Medicine, Fuzhou, Fujian, PR China

☉ These authors contributed equally to this work.
* chenliangwan@fjmu.edu.cn

## Abstract

### Background

Preoperative inflammatory biomarker ratios to predict adverse outcomes in patients with acute type A aortic dissection (AAD) were assessed in this study, and a prognostic nomogram to guide anti-inflammatory therapy was developed..

### Methods

We retrospectively analyzed 673 adult AAD patients who underwent surgery. Preoperative hematological parameters, including neutrophil, lymphocyte, and platelet counts; hemoglobin (Hb) and albumin levels; and composite indices including the neutrophil–lymphocyte ratio (NLR), platelet–lymphocyte ratio, neutrophil–platelet ratio, and platelet–albumin ratio, were evaluated. The univariate and multivariate logistic regression identified in-hospital mortality predictors, and a nomogram was constructed. The internal validation included bootstrapping with discrimination assessed by the C-index and calibration by the Hosmer–Lemeshow test.

### Results

The univariate analysis revealed Hb, D-dimer, blood urea nitrogen, and albumin levels; the NLR; the aortic root concomitant procedure; ventilation support time and multiple organ dysfunction syndrome (MODS) as perioperative mortality predictors; after multivariate adjustment, decreased Hb level, elevated NLR, and the presence of MODS independently predicted in-hospital mortality. The nomogram that integrated these predictors achieved a corrected C-index of 0.846 and an area under the curve

**Data availability statement:** All relevant data are within the manuscript and its Supporting Information files.

**Funding:** This work was supported by the following grants: L.X.C was funded by Joint Funds for the innovation of science and Technology, Fujian province (No. 2023Y9155). L.X.C was funded by Fujian Provincial Natural Science Foundation of China (NO.2024J01123691). C.L.W received grant from Fujian Provincial Center for Cardiovascular Medicine Construction Project (NO.2128300201). The funders had no role in study design, data collection and analysis, decision to publish, or preparation of the manuscript.

**Competing interests:** The authors have declared that no competing interests exist.

of 0.843, which demonstrated strong calibration and a Hosmer–Lemeshow P = 0.91. At the optimal probability cutoff of 0.124, the sensitivity was 77.2%, the specificity was 78.2%, and the accuracy was 78.1%.

## Conclusion

The NLR and preoperative Hb level, combined with postoperative MODS, independently predict in-hospital death in patients with AAD. Additionally, a nomogram combining these factors accurately predicts short-term mortality and aids in the personalized risk assessment and may assist in improving the prognosis.

## Introduction

Acute type A aortic dissection (AAD) is a life-threatening cardiovascular condition that is associated with high mortality rates [1,2]. Timely intervention is generally required to avoid catastrophic outcomes. To provide a vital lifeline for patients experiencing this condition, surgical procedures are often needed [3]. Unfortunately, studies have shown that the incidence of postoperative multiple organ dysfunction syndrome (MODS) in patients with AAD is close to 30% [4], and the in-hospital mortality rate is greater than 20% [5]. The early detection of high-risk patients and management of related risk factors are critical for improving the clinical outcome of AAD patients.

Systemic inflammatory responses triggered by AAD, cardiopulmonary bypass, or anesthesia are likely common in AAD patients [6,7]. Inflammation pervades the occurrence, development and progression of AAD and strongly affects the prognosis [8]. Previous studies have shown that some inflammation-related biomarkers are risk factors for postoperative mortality in patients with AAD [9–11]. Additionally, serum albumin, which is associated with systemic inflammation, thrombosis, and oxidative stress, has been widely studied as a predictor of poor prognosis of aortic dissection [12–14].

Recently, many complex systemic inflammatory markers, such as the neutrophil-to-lymphocyte ratio, platelet-to-lymphocyte ratio, and nitrogen-to-albumin ratio, have been used to predict the prognosis of AAD [15–18]. These measures are correlated with AAD outcomes, but their reliability as prognostic predictors remains unclear. Thus, further evidence is needed to better understand the complex relationships between these indicators and adverse clinical outcomes. This study assessed the predictive value of preoperative inflammatory biomarker ratios for adverse postoperative outcomes in AAD patients to construct a prognostic nomogram model and guide anti-inflammatory treatment.

## Methods

### Study population and ethical approval

A retrospective analysis was performed on 673 patients who underwent surgery for AAD at Fujian Medical University Union Hospital from January 2015 to December 2022. A diagnosis of AAD was made using either aortic computed tomography

angiography or magnetic resonance imaging. Patients 18 years of age or older underwent aortic surgery for AAD. The exclusion criteria were as follows: missing data, recurrent AAD, a history of cardiac surgery, severe infectious diseases and chronic organ dysfunction, pregnancy, and autoimmune diseases. This study was approved by the Fujian Medical University Union Hospital Ethics Committee (No. 2023KY253) and strictly complied with the Declaration of Helsinki. The requirement for informed consent was waived because it was a retrospective observational study. The study title has been revised for clarity, but the research population, data sources, and objectives remain consistent with the original ethics approval.

## Surgical procedure

Surgery was performed under general anesthesia and with the support of cardiopulmonary bypass under hypothermic circulatory arrest. The specific surgical methods included reconstruction of the aortic root, replacement of the ascending aorta, partial aortic arch replacement and stent grafting.

## Data collection

Data were accessed from the hospital's medical records system from June 21, 2024, to November 10, 2024. The collected variables included medical history, risk factors, baseline characteristics, surgical details, multiple organ dysfunction syndrome ((MODS), in-hospital and 30-day mortality, and preoperative blood biomarkers, which included platelet, white blood cell, neutrophil, and lymphocyte counts; and albumin, Hb, D-dimer, fibrinogen, aspartate transaminase, and alanine transaminase levels. In contrast with single inflammatory biomarkers, composite indices derived from routine hematological parameters offer enhanced prognostic value. In this study, the following four systemic immune inflammation indices were evaluated: the neutrophil–lymphocyte ratio (NLR), the platelet–lymphocyte ratio (PLR), the neutrophil–platelet ratio (NPR), and the platelet–albumin ratio (PAR).

## Statistical analysis

All statistical analyses and graphical visualizations in this study were performed using R (V4.3.1) statistical software. Patients were divided into survival and death groups on the basis of their survival status during hospitalization. Non-normally distributed quantitative data are presented as the median (Q1, Q3), and the Mann–Whitney U test was used for intergroup comparisons. Normally distributed quantitative data are presented as the mean and standard deviation (mean ± standard deviation), and a t test was used for intergroup comparisons. Categorical data are presented as numbers and percentages (%), and the chi-square test was used for intergroup comparisons. The "Table 1" package was used to integrate the analysis results. Clinical indicators were initially screened using univariate logistic regression analysis. Variables for which P was < 0.05 in the univariate analysis were included in the multivariate logistic regression. To avoid multicollinearity, blood cell parameters that were used to calculate the systemic immune inflammation index were excluded. Clinical indicators that retained statistical significance (P < 0.05) in the multivariate model were identified as predictors of in-hospital mortality. The detection of multicollinearity was conducted using the variance inflation factor (VIF). If VIF < 5, the collinearity between variables is minimal and acceptable. When 5 ≤ VIF < 10 (indicating moderate collinearity) and VIF ≥ 10 (indicating severe multicollinearity), the variables are removed. A nomogram was constructed on the basis of the final multivariate logistic regression model derived from the entire cohort.

## Measurement of performance and validation

The modeling dataset included 673 patients with AAD who underwent surgical intervention at Fujian Medical University Union Hospital between January 2015 and December 2022. A logistic regression model was developed and its internal validation was performed using the bootstrap resampling method with replacement. Then, the model performance was

**Table 1. Patient characteristics, procedural data, and outcomes in the two patient groups.**

| Variables | Non-in-hospital mortality group (n = 616) | In-hospital mortality group (n = 57) | P-value |
|---|---|---|---|
| Gender(female), n(%) | 140 (22.7%) | 17 (29.8%) | 0.294 |
| Age(year), Mean(SD) | 51.96 (11.57) | 53.65 (10.69) | 0.261 |
| BMI(Kg/m$^2$), Mean(SD) | 25.24 (3.46) | 25.49 (3.09) | 0.567 |
| Hypertension, n(%) | 416 (67.5%) | 38 (66.7%) | 1.000 |
| History of stroke, n(%) | 30 (4.9%) | 5 (8.8%) | 0.338 |
| Coronary artery disease, n(%) | 54 (8.8%) | 7 (12.3%) | 0.520 |
| Diabetes, n(%) | 35 (5.7%) | 3 (5.3%) | 1.000 |
| Chronic obstructive pulmonary disease, n(%) | 16 (2.6%) | 3 (5.3%) | 0.457 |
| Smoking, n(%) | 265 (43.0%) | 23 (40.4%) | 0.803 |
| Drinking, n (%) | 142 (23.1%) | 16 (28.1%) | 0.489 |
| White blood cell(×10$^9$/L), Mean(SD) | 12.45 (4.32) | 14.72 (5.61) | 0.004 |
| Neutrophil(×10$^9$/L), Mean (SD) | 10.47 (4.13) | 12.07 (3.66) | 0.003 |
| Fibrinogen(g/l), Mean(SD) | 3.40 (1.75) | 3.08 (1.75) | 0.193 |
| D-dimer(ug/ml), Median(Q1,Q2) | 10.04 (3.04, 10.04) | 20.00 (11.18, 20.00) | <0.001 |
| Blood urea nitrogen(µmol/L), Median(Q1,Q2) | 6.20 (4.90, 6.20) | 7.70 (5.90, 7.70) | 0.014 |
| Creatinine(µmol/L), Median(Q1,Q2) | 83.15 (67.00, 83.15) | 92.00 (71.00, 92.00) | 0.116 |
| AST(IU/L), Median(Q1,Q2) | 27.00 (20.00, 27.00) | 34.00 (23.00, 34.00) | 0.460 |
| ALT(IU/L), Median(Q1,Q2) | 27.00 (18.00, 27.00) | 35.00 (23.00, 35.00) | 0.403 |
| Hemoglobin(g/l), Mean(SD) | 129.92 (19.57) | 123.19 (19.36) | 0.015 |
| Lymphocyte(×10$^9$/L), Mean(SD) | 1.13 (0.75) | 1.07 (0.62) | 0.500 |
| Platelet(×10$^9$/L), Mean(SD) | 185.35 (74.09) | 179.14 (80.19) | 0.575 |
| Albumin(g/l), Mean(SD) | 37.11 (4.91) | 35.68 (4.54) | 0.027 |
| Aortic root concomitant procedure | | | |
| Bentall, n(%) | 98 (15.9%) | 10 (17.5%) | 0.894 |
| Carbol, n(%) | 8 (1.3%) | 0 (0%) | 0.821 |
| David, n(%) | 11 (1.8%) | 1 (1.8%) | 0.688 |
| Wheat, n(%) | 1 (0.2%) | 0 (0%) | 0.932 |
| Aortic valve replacement, n(%) | 19 (3.1%) | 2 (3.5%) | 1.000 |
| Aortic sinus repaired, n(%) | 220 (35.7%) | 27 (47.4%) | 0.109 |
| CABG, n(%) | 51 (8.3%) | 9 (15.8%) | 0.097 |
| Operative time(min), Mean(SD) | 295.09 (59.65) | 333.86 (87.17) | 0.002 |
| Aortic cross-clamp time(min), Mean(SD) | 52.96 (21.13) | 58.00 (24.22) | 0.134 |
| Cardiopulmonary bypass time(min), Mean(SD) | 143.26 (36.93) | 167.44 (64.77) | 0.007 |
| Ventilation support time(h), Median (Q1,Q2) | 44.00 (24.00, 44.00) | 154.00 (40.00, 154.00) | <0.001 |
| ICU stay time(d), Median(Q1,Q2) | 4.00 (2.79, 4.00) | 8.00 (2.00, 8.00) | 0.011 |
| Length of hospital stay(d), Median(Q1,Q2) | 19.00 (15.00, 19.00) | 12.00 (4.00, 12.00) | 0.128 |
| MODS, n(%) | 141 (22.9%) | 37 (64.9%) | <0.001 |

BMI, body mass index; ALT, alanine aminotransferase; AST, aspartate aminotransferase; MODS, multiple organ dysfunction syndrome; CABG, coronary artery bypass grafting.

p-value less than 0.05, indicating a statistically significant difference between groups.

assessed via discrimination and calibration. Discrimination is defined as the probability that a model can correctly distinguish between nonevents and events. In this study, a C-index was used for discrimination. Calibration is a measurement of how closely the predicted probabilities agree with the actual outcomes. Calibration was evaluated via the Hosmer–Lemeshow test (good fit: P > 0.05) and calibration plots. The optimal cutoff value was determined on the basis of the receiver operating characteristic curve analysis of the modeling dataset to determine the best combination of Youden's index and clinical practicality.

## Results

### Clinical baseline data

A total of 673 patients with aortic dissection were included in this study, 57 of whom died during hospitalization. The study selection process is shown in Fig 1. All patient data, including demographic and clinical data, are presented in Table 1.

### Univariate and multivariate logistic regression analysis

The univariate logistic regression revealed several perioperative predictors of in-hospital mortality in AAD patients as follows: Hb level (OR=0.98; 95% CI 0.97–1.00; P = 0.014), D-dimer level (OR=1.10; 95% CI 1.05–1.15; P < 0.001), blood urea nitrogen level (OR=1.06; 95% CI 1.01–1.11; P = 0.011), serum albumin level (OR=0.94; 95% CI 0.89–1.00; P = 0.035), NLR (OR=1.12; 95% CI 1.09–1.16; P < 0.001), aortic root concomitant procedure (OR=1.85; 95% CI 1.01–3.36; P = 0.045), ventilation support time (OR=1.00; 95% CI 1.00–1.00; P < 0.001), and MODS (OR=6.23; 95% CI 3.51–11.08; P < 0.001). To avoid multicollinearity, the blood cell parameters used to calculate the systemic immune-inflammation index were excluded. After multivariate adjustment, only Hb level (OR=0.98, 95% CI 0.96–0.99; P = 0.006), NLR (OR=1.11, 95% CI 1.07–1.15; P < 0.001) and MODS (OR=3.64, 95% CI 1.77–7.50; P < 0.001) remained independent predictors of in-hospital mortality. D-dimer, blood urea nitrogen, and albumin levels; aortic root concomitant procedure; and ventilation support time were not retained as significant predictors in the multivariable model (Table 2).

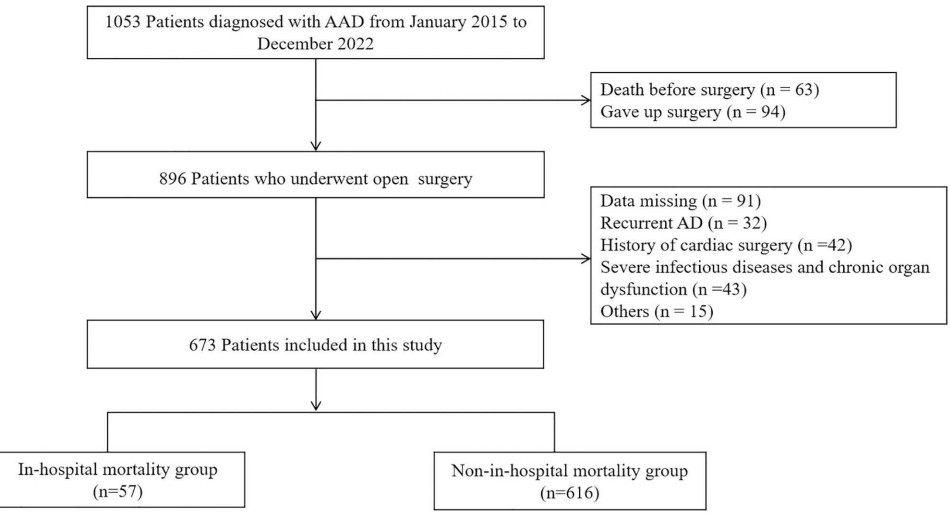

**Fig 1. Study selection process.**

**Table 2. Univariate and multivariate logistic regression analyses of all patients.**

| Characteristics | Univariate | | Multivariate | | VIF |
|---|---|---|---|---|---|
| | OR (95%CI) | P-value | OR (95%CI) | P-value | |
| Age (year) | 1.01 (0.99-1.04) | 0.289 | | | |
| BMI (kg/m²) | 1.02 (0.94-1.10) | 0.601 | | | |
| Gender | 0.69 (0.38-1.26) | 0.228 | | | |
| Hypertension | 0.96 (0.54-1.71) | 0.894 | | | |
| Stroke | 1.88 (0.70-5.05) | 0.211 | | | |
| Coronary artery disease | 1.46 (0.63-3.37) | 0.379 | | | |
| Diabetes | 0.92 (0.27-3.10) | 0.896 | | | |
| Chronic obstructive pulmonary disease | 2.08 (0.59-7.38) | 0.255 | | | |
| Somking | 0.90 (0.52-1.56) | 0.697 | | | |
| Drinking | 1.30 (0.71-2.39) | 0.394 | | | |
| White blood cell (×10⁹/L) | 1.09 (1.04-1.15) | <0.001 | | | |
| Neutrophil (×10⁹/L) | 1.09 (1.03-1.15) | 0.005 | | | |
| Hemoglobin (g/l) | 0.98 (0.97-1.00) | 0.014 | 0.98 (0.96-0.99) | 0.006 | 1.399 |
| Lymphocyte (×10⁹/L) | 0.88 (0.56-1.37) | 0.561 | | | |
| Platelet (×10⁹/L) | 1.00 (1.00-1.00) | 0.547 | | | |
| Fibrinogen (g/l) | 0.89 (0.75-1.06) | 0.191 | | | |
| D-dimer (ug/ml) | 1.10 (1.05-1.15) | <0.001 | 1.03 (0.98-1.09) | 0.218 | 1.268 |
| Blood urea nitrogen (μmol/L) | 1.06 (1.01-1.11) | 0.011 | 0.97 (0.89-1.05) | 0.398 | 1.285 |
| Creatinine (μmol/L) | 1.00 (1.00-1.00) | 0.06 | | | |
| AST (IU/L) | 1.00 (1.00-1.00) | 0.707 | | | |
| ALT (IU/L) | 1.00 (1.00-1.00) | 0.534 | | | |
| Albumin (g/l) | 0.94 (0.89-1.00) | 0.035 | 0.99 (0.92-1.07) | 0.787 | 1.410 |
| SII | 1.00 (1.00-1.00) | 0.163 | | | |
| NLR | 1.12 (1.09-1.16) | <0.001 | 1.11 (1.07-1.15) | <0.001 | 1.121 |
| PLR | 1.00 (1.00-1.00) | 0.875 | | | |
| NPR | 0.97 (0.95-1.00) | 0.033 | | | |
| PAR | 0.99 (0.88-1.11) | 0.862 | | | |
| Aortic root concomitant procedure | 1.85 (1.01-3.36) | 0.045 | 1.87 (0.93-3.73) | 0.078 | 1.085 |
| CABG | 2.08 (0.96-4.47) | 0.062 | | | |
| Aortic cross-clamping time (min) | 1.01 (1.00-1.02) | 0.091 | | | |
| Ventilation support time (h) | 1.00 (1.00-1.00) | <0.001 | 1.00 (1.00-1.00) | 0.119 | 1.233 |
| MODS | 6.23 (3.51-11.08) | <0.001 | 3.64 (1.77-7.50) | <0.001 | 1.368 |

BMI, body mass index; AST, aspartate aminotransferase; ALT, alanine aminotransferase; ALB, albumin. SII, systemic immune inflammation index; NLR, neutrophil-lymphocyte ratio; PLR, platelet-lymphocyte ratio; NPR, neutrophil- platelet; PAR, platelet-albumin. MODS, multiple organ dysfunction syndrome; CABG, coronary artery bypass grafting.

p-value less than 0.05, indicating a statistically significant difference.

## Construction of the nomogram prediction model

The multivariate analysis confirmed the Hb level, NLR, and MODS as independent predictors of in-hospital mortality. The prediction model was operationalized through a nomogram (Fig 2), which translates predictor values into mortality probabilities. The final logistic model for in-hospital mortality was defined by the following equation: $\text{Logit}(P) = -1.9917 - 0.0199 \times Hb + 0.1103 \times NLR + 1.4995 \times MODS$ (yes = 1; no = 0).

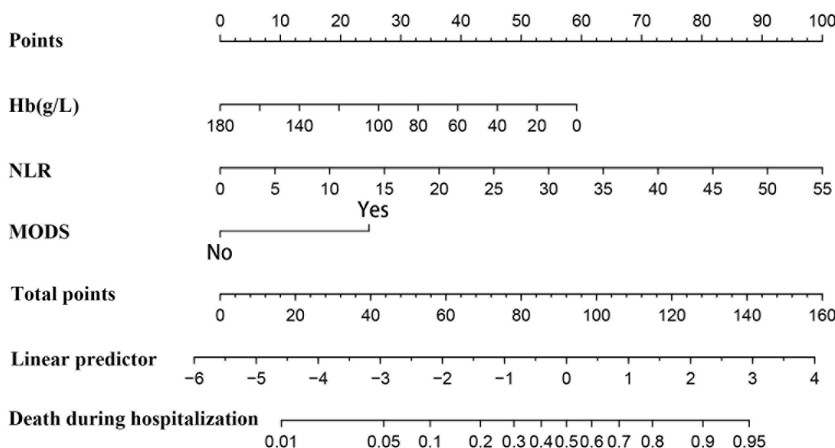

**Fig 2. A nomogram for predicting in-hospital mortality in AAD patients.** Predictors: hemoglobin (Hb) concentration (g/L), neutrophil-to-lymphocyte ratio (NLR), and multiorgan dysfunction score (MODS). The total points translate to the predicted mortality probability (0.01–0.95).

## Performance and validation of the nomogram

Internal validation was performed using 1000 bootstrap resamples of the modeling cohort. Results demonstrated good discrimination and calibration of the model in predicting in-hospital mortality risk for AAD patients (corrected C-index: 0.846; 95% CI: 0.795–0.893). The Hosmer–Lemeshow test ($\chi^2 = 3.4$, P = 0.91) and calibration plot indicated excellent agreement between the predicted and observed in-hospital mortality risks. Furthermore, the calibration curve closely aligned with the ideal line, which confirmed satisfactory model calibration (Fig 3). The receiver operating characteristic (ROC) curve analysis yielded an area under the curve (AUC) of 0.843 (95% CI: 0.793–0.893) (Fig 4) and indicated robust predictive accuracy. The optimal probability cutoff that was derived from the ROC curve in the derivation cohort was 0.124 with corresponding sensitivity, specificity, and overall accuracy values of 77.2%, 78.2%, and 78.1%, respectively.

## Discussion

Although the mortality rate of AAD after surgery has decreased in recent years it is still high because of numerous post-operative complications. Preoperative hematological inflammatory indicator ratios are closely related to AAD prognosis [15,16,18,19]; nevertheless, most studies are limited to individual inflammatory markers and lack comprehensive assessment measures. In this study, a nomogram model integrating plasma composite inflammatory markers and multiorgan dysfunction was constructed, evaluated, and validated. Overall, the predicted probabilities of the model were in good agreement with the actual probabilities, and the model provides an important risk assessment tool for in-hospital mortality after surgery in patients with AAD.

Our current report presents valuable information and adds to the literature. Previous studies focused on single hematological inflammatory indicators and a single-organ adverse event as the main endpoint while lacking systematic comorbidity data. In contrast, our current study not only included an assessment of comorbidity status across multiple organs but also investigated the ratios of multiple blood inflammatory indicators and clinical risk factors, which enhances the reliability of our study results and the usability of the model.

On the basis of our findings, the Hb level is a significant independent predictor of in-hospital mortality in patients with AAD, which is consistent with the established literature. Our predictive model that incorporates the Hb level demonstrated robust discriminatory capacity and reinforces the critical role of hemodynamic and inflammatory pathways in AAD outcomes [20]. This finding aligns with prior research showing that lower Hb levels reflect compromised oxygen-carrying

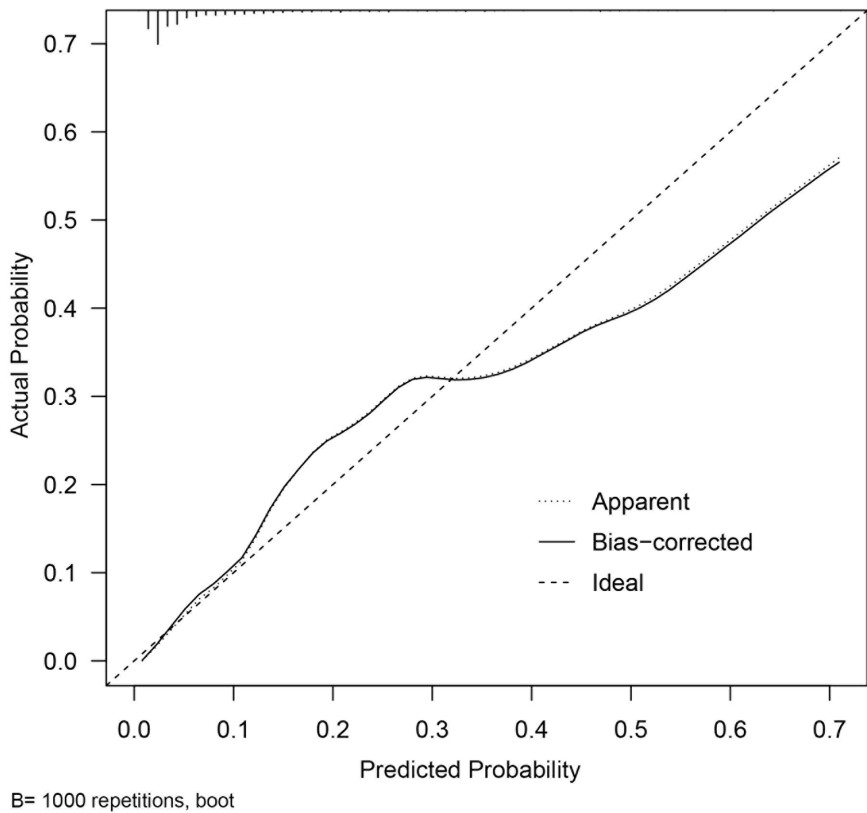

B= 1000 repetitions, boot

**Fig 3. A calibration curve of the mortality prediction model.** A comparison of the predicted vs. observed in-hospital mortality probabilities is shown as well as the apparent (solid), bootstrap-corrected (dashed), and ideal (dotted) calibration curves.

capacity, which potentially exacerbates end-organ malperfusion during dissection. The inclusion of Hb level in our model parallels its use in validated risk stratification tools, which also identified Hb level as a key prognostic variable for AAD mortality [21]. Notably, similar Hb–mortality associations have been documented in acute type B aortic dissection and suggest broader pathophysiological relevance across aortic syndromes [22,23]. Furthermore, anemia may reflect a significant hemorrhage, from either the dissection itself or malperfusion complications, which leads to reduced oxygen-carrying capacity and exacerbated end-organ ischemia. Anemia has also been linked to impaired coagulation and increased inflammatory burden in patients experiencing cardiovascular emergencies, which potentially contributes to a vicious cycle of deterioration [22,24]. Our results underscore the critical importance of the early recognition and management of anemia in AAD patients.

The NLR represents a potent independent predictor of cardiovascular outcomes [25]. This readily calculable ratio serves as a robust biomarker that reflects a systemic inflammatory status and physiological stress responses [26,27]. Elevated NLR values indicate pronounced neutrophilia coupled with relative lymphopenia, which indicates acute inflammatory states with tissue damage alongside stress-induced immunosuppression or lymphocyte apoptosis [26]. Such dysregulated inflammation constitutes a fundamental characteristic of AAD pathophysiology [25] and promotes endothelial dysfunction, coagulopathy, and progression to multiorgan failure. These observations align with established research that demonstrates associations between an elevated NLR and adverse clinical outcomes across aortic syndromes and critical cardiovascular conditions [28].

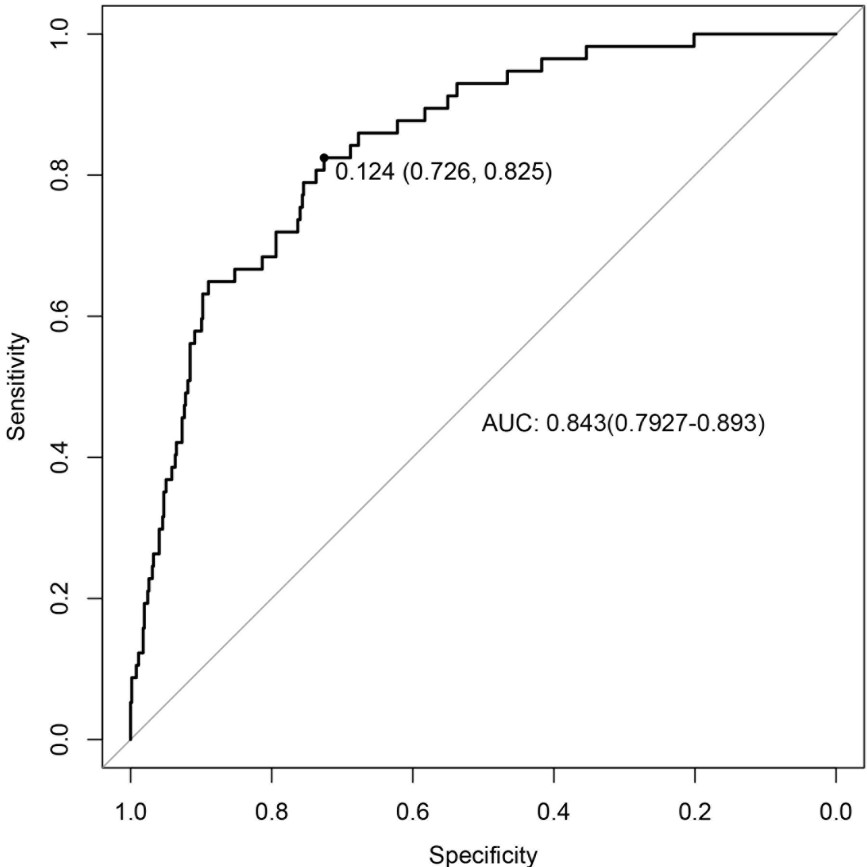

**Fig 4. A receiver operating characteristic (ROC) curve.** The discriminative ability of the model with AUC = 0 for predicting in-hospital mortality was 0.843 (95% CI: 0.793–0.893). AUC represents the area under the curve.

The strong predictive value of the MODS score emphasizes the critical role of end-organ damage in survival outcomes [29]. AAD often triggers severe malperfusion syndromes involving the brain, kidneys, mesentery, limbs, and spinal cord [30]. The emergence or progression of MODS reflects the cumulative effects of ischemia–reperfusion injury, systemic inflammation, and subsequent complications such as acute kidney injury or liver failure [31]. As a quantifiable measure of this systemic dysfunction, the MODS score demonstrated significant predictive accuracy in our model. This highlights how mortality risk is closely associated with the severity of concomitant organ compromise, which extends beyond the initial aortic injury itself [4].

By integrating the Hb level, the NLR, and MODS into a predictive model, we provide a tool that leverages routine clinical data that is available early in the disease course of the patient. This model can provide advantages over risk scores that require complex calculations or less commonly available parameters. A major strength of this model lies in the self-contained nature of the predictors; for example, the Hb level and the components of the NLR are derived from routine complete blood counts that are performed ubiquitously upon patient admission, whereas MODS assessment relies on established and widely used clinical criteria readily evaluated during initial disease management and ICU care of the patient. This makes the model readily applicable at the point of care without the need for specialized or delayed testing and enables rapid risk stratification to guide urgent clinical decision-making. Our model exhibits strong discriminative power (AUC = 0.843) and addresses distinct pathophysiological dimensions that are relevant to the complex biology of

AAD, including blood loss/oxygen delivery, systemic inflammation, and end-organ dysfunction, which results in enhanced accuracy.

In addition, the internal validation with 1000 bootstrap resamples confirmed the robust performance of our AAD in-hospital mortality prediction model, which demonstrated excellent discrimination and calibration. Calibration plots showed close alignment between the predicted and observed outcomes. At the clinically relevant threshold of 12.4%, the model achieved balanced diagnostic accuracy, which supports its utility for acute clinical risk stratification in AAD patients.

## Limitations

Limitations of this study include a limited generalizability inherent in its retrospective, single-center nature and the absence of external validation for the constructed model. Prospective multicenter validation in future research is essential to strengthen its reliability and confirm wider applicability.

## Conclusion

Preoperative Hb level, the NLR and postoperative concurrent MODS are independent influencing factors for in-hospital mortality in AAD patients. A nomogram constructed on the basis of the above factors has high predictive value for short-term postoperative mortality. This nomogram model integrates preoperative hematological inflammatory indicator ratios and postoperative concurrent MODS, which can help clinicians conduct more accurate multiple-indicator joint individualized prognosis predictions. Targeted treatment decisions can be made according to the risk situation of each patient, which is conducive to promoting individualized diagnosis and treatment and implementing reasonable treatment plans for AAD patients, which thereby may assists in improving patient outcomes.

## Supporting information

**S1 Data.**
(XLSX)

## Acknowledgments

Not applicable.

## Author contributions

**Conceptualization:** Lei Wang.

**Data curation:** Min-xia Xie.

**Formal analysis:** Lin-feng Xie.

**Funding acquisition:** Liang-wan Chen.

**Investigation:** Lin-feng Xie, Min-xia Xie.

**Methodology:** Xiao-Chai Lv, Lin-feng Xie, Yan-ting Hou.

**Software:** Min-xia Xie.

**Supervision:** Liang-wan Chen.

**Validation:** Xiao-Chai Lv, Liang-wan Chen.

**Visualization:** Lei Wang.

**Writing – original draft:** Xiao-Chai Lv.

**Writing – review & editing:** Liang-wan Chen.

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
