## [Decision Letter · Decision Letter 0]

30 Apr 2025

Dear Dr. chen,

Thank you for submitting your manuscript to PLOS ONE. After careful consideration, we feel that it has merit but does not fully meet PLOS ONE’s publication criteria as it currently stands. Therefore, we invite you to submit a revised version of the manuscript that addresses the points raised during the review process.

We look forward to receiving your revised manuscript.

Kind regards,

Tomasz W. Kaminski

Academic Editor

PLOS ONE

“This work was supported by the following grants: L.X.C was funded by Joint Funds for the innovation of science and Technology, Fujian province (No. 2023Y9155). L.X.C was funded by Fujian Provincial Natural Science Foundation of China  (NO.2024J01123691). C.L.W received grant from Fujian Provincial Center for Cardiovascular Medicine Construction Project (NO.2128300201).”

Additional Editor Comments:

Dear Authors,

Thank you for submitting your manuscript. After careful evaluation, we have received two sets of reviewer comments, and both reviewers recommend major revision.

Key issues raised include:

Concerns about the statistical analysis, specifically the use of discretized variables instead of continuous ones, lack of multicollinearity assessment, unclear significance thresholds and justification for the 7:3 training/validation split.

Questions regarding biomarker selection criteria (AUC > 0.6), unclear reporting of the postoperative ventilation time cutoff, potential multicollinearity between biomarkers and the need for a more thorough discussion of study limitations (retrospective, single-center design) and external validation.

We ask that you address these points thoroughly in your revised submission. Please provide detailed responses to each comment and make the necessary changes to the manuscript.

We look forward to receiving your revised work.

Best regards,

Tomasz W. Kaminski

Reviewers' comments:

Reviewer's Responses to Questions

**Comments to the Author**

1. Is the manuscript technically sound, and do the data support the conclusions?

Reviewer #1: Partly

Reviewer #2: Partly

2. Has the statistical analysis been performed appropriately and rigorously?

Reviewer #1: Yes

Reviewer #2: Yes

3. Have the authors made all data underlying the findings in their manuscript fully available?

Reviewer #1: Yes

Reviewer #2: Yes

4. Is the manuscript presented in an intelligible fashion and written in standard English?

Reviewer #1: Yes

Reviewer #2: Yes

Reviewer #1: This manuscript aims to evaluate the predictive value of preoperative inflammatory biomarker ratios for adverse postoperative outcomes in AAD patients and to construct a prognostic nomogram model to guide anti-inflammatory treatment. My comments on statistical analysis part are given below.

1. Why not use continuous variable directly in the analysis? Logistic regression handles continuous variable very well. Discretization (e.g. use cut-off point to group continuous values into two groups) will discard information contained in the original continuous variable and thus will lose efficiency in, say, prediction.

2. Correlation between covariates should be examined. In case of co-linearity, one needs to keep one of the co-linear pair. These steps will influence the final results and conclusions.

3. In multivariate Logistic regression, interactions may be included.

4. Please explain why some tests use 0.05 as the significance level while others use 0.1.

5. Please explain why $7:3$ is used as the training sample to validation sample ratio. Will such a ratio cause over-fitting problem?

Reviewer #2: The manuscript addresses an important clinical issue with relevant biomarker analyses. However, several aspects require improvement. Firstly, the selection criteria for biomarkers appear arbitrary (AUC > 0.6), lacking robust justification. Clearly define the rationale behind this threshold or consider adjusting based on clinical relevance. Secondly, the postoperative ventilation cutoff (86.5 min) seems unusually brief clinically; clarify if units were misreported (minutes vs hours). Additionally, multicollinearity between biomarkers such as WBC/PLT and Ne/PLT is possible; include correlation analysis or justify their simultaneous use. Lastly, discuss potential biases due to retrospective single-center design and validate externally if feasible. Recommendation: revise.

**Do you want your identity to be public for this peer review?** For information about this choice, including consent withdrawal, please see our Privacy Policy

Reviewer #1: No

Reviewer #2: **Yes: ** Robert J. Chen, MD, MPH

---

## [Author Response · Author response to Decision Letter 1]

15 May 2025

Dear Editors and Reviewers: 

Thank you for your letter and for the reviewers “Prognostic nomogram integrated with inflammatory marker ratios for assessing in-hospital mortality risk in patients with acute type A aortic dissection” (PONE-D-25-13134). Those comments are all valuable and very helpful for 

revising and improving our paper. We have studied comments carefully and the responds to the reviewer’s comments are as following:

Response: Thank you for your feedback. We have carefully revised the manuscript to ensure full compliance with PLOS ONE’s style requirements. Please let us know if further adjustments are needed.

2. Thank you for stating the following financial disclosure: “This work was supported by the following grants: L.X.C was funded by Joint Funds for the innovation of science and Technology, Fujian province (No. 2023Y9155). L.X.C was funded by Fujian Provincial Natural Science Foundation of China (NO.2024J01123691). C.L.W received grant from Fujian Provincial Center for Cardiovascular Medicine Construction Project (NO.2128300201).”

Response: Thank you for your comment. We confirm that the funders had no role in the study design, data collection and analysis, decision to publish, or preparation of the manuscript. The funding statements in the cover letter and manuscript have been amended accordingly.

Response: Thank you for your note. We confirm that all raw data required to replicate the study findings are included in the manuscript. The minimum data set is available in the following file: data.xlsx, and it has been uploaded as a Supporting Information file. We have updated the Data Availability Statement in the manuscript to: "All the data are within the manuscript and its Supporting information files." Please let us know if further clarification is needed.

Response: I would like to thank the reviewers for their valuable advice. We have removed ethics statements from other sections besides the Methods.

Additional Editor Comments:

Thank you for submitting your manuscript. After careful evaluation, we have received two sets of reviewer comments, and both reviewers recommend major revision.

Key issues raised include:

1.Concerns about the statistical analysis, specifically the use of discretized variables instead of continuous ones, lack of multicollinearity assessment, unclear significance thresholds and justification for the 7:3 training/validation split.

Response: Thank you for the reviewers' comments. The reviewers provided professional opinions on statistical analysis, which were helpful for the improvement of our research. First, regarding the assessment of multicollinearity, according to the conventional analysis process, all variables entering the multifactor analysis can only be qualified if they have undergone multicollinearity testing. At the beginning of writing the article, our team conducted an assessment of multicollinearity, and the collinearity of all variables entering the multifactor analysis was acceptable. We considered this to be part of the regular process, so we did not report on this content. We understand the concerns of the reviewers and editors, and therefore we supplemented this content in the statistical analysis, results, and discussion sections. Secondly, the 7:3 ratio is the empirical segmentation adopted by most clinical research prediction model studies, so we referred to this empirical ratio. Thank you for your comments.

2.Questions regarding biomarker selection criteria (AUC > 0.6), unclear reporting of the postoperative ventilation time cutoff, potential multicollinearity between biomarkers and the need for a more thorough discussion of study limitations (retrospective, single-center design) and external validation.

Response: Thank you for the reviewer's comments, which are helpful for the improvement of our research. As can be seen from the article, this is a process of gradually screening variables. In the first round of screening, we used ROC to screen all biomarkers with the aim of selecting variables that are relatively related to the outcome, as well as choosing the best cutoff value, thus selecting the value of 0.6. This is because it is not very strict but still has distinguishing ability. On the other hand, the limitations of the first edition may not be satisfactory, so we have re-discussed the limitations of this study. Thank you for your comments.

3.We ask that you address these points thoroughly in your revised submission. Please provide detailed responses to each comment and make the necessary changes to the manuscript.

Response: Thank you very much for your comments. We will address each point thoroughly in our revised submission and provide detailed responses to each comment, along with the necessary changes to the manuscript.

Reviewers' comments:

1. Is the manuscript technically sound, and do the data support the conclusions?

Reviewer #1: Partly

Response #1:Thank you for the reviewers' comments.

Reviewer #2: Partly

Response #2:Thank you for the reviewers' comments.

2. Has the statistical analysis been performed appropriately and rigorously?

Reviewer #1: Yes

Response #1: Thanks

Reviewer #2: Yes

Response #2: Thanks

3. Have the authors made all data underlying the findings in their manuscript fully available?

Reviewer #1: Yes

Response #1: Thanks

Reviewer #2: Yes

Response #2: Thanks

4. Is the manuscript presented in an intelligible fashion and written in standard English?

Reviewer #1: Yes

Response #1: Thanks

Reviewer #2: Yes

Response #2: Thanks

5. Review Comments to the Author

Reviewer #1: This manuscript aims to evaluate the predictive value of preoperative inflammatory biomarker ratios for adverse postoperative outcomes in AAD patients and to construct a prognostic nomogram model to guide anti-inflammatory treatment. My comments on statistical analysis part are given below.

(1)Why not use continuous variable directly in the analysis? Logistic regression handles continuous variable very well. Discretization (e.g. use cut-off point to group continuous values into two groups) will discard information contained in the original continuous variable and thus will lose efficiency in, say, prediction.

Response #1: We appreciate the reviewers' comments, which are very professional. After discussion, our team believes that it is necessary to maintain the original results, as the ultimate goal of this study is clinical application. A specific value as the basis for grouping is more convenient in clinical practice. Thank you for your comments.

(2) Correlation between covariates should be examined. In case of co-linearity, one needs to keep one of the co-linear pair. These steps will influence the final results and conclusions.

Response #1: Thank you for the reviewers' comments. The reviewers provided professional opinions on statistical analysis, which were helpful for the improvement of our research. First, regarding the assessment of multicollinearity, according to the conventional analysis process, all variables entering the multifactor analysis can only be qualified if they have undergone multicollinearity testing. At the beginning of writing the article, our team conducted an assessment of multicollinearity, and the collinearity of all variables entering the multifactor analysis was acceptable. We considered this to be part of the regular process, so we did not report on this content. We understand the concerns of the reviewers and editors, and therefore we supplemented this content in the statistical analysis, results, and discussion sections. Thank you for your comments.

(3) In multivariate Logistic regression, interactions may be included.

Response #1: Thank you for the reviewers' comments.We understand the concerns of the reviewers, and therefore we supplemented the results of the collinearity test.

(4) Please explain why some tests use 0.05 as the significance level while others use 0.1.

Response #1: We appreciate the reviewer's insightful question regarding the rationale behind varying significance thresholds (α=0.05 vs. α=0.1). In our study, a significance level of α=0.1 was selected for specific analyses involving small-to-moderate sample sizes to mitigate the risk of Type II errors (i.e., failing to detect true effects due to limited statistical power). This approach aligns with methodological recommendations for exploratory analyses or pilot studies, where overly stringent thresholds (e.g., α=0.05) may overlook clinically meaningful associations in resource-constrained settings. The model we constructed using 0.1 threshold also demonstrated good discrimination. Thank you for your comments.

5. Please explain why $7:3$ is used as the training sample to validation sample ratio. Will such a ratio cause over-fitting problem?

Reviewer #2: The manuscript addresses an important clinical issue with relevant biomarker analyses. However, several aspects require improvement. Firstly, the selection criteria for biomarkers appear arbitrary (AUC > 0.6), lacking robust justification. Clearly define the rationale behind this threshold or consider adjusting based on clinical relevance. Secondly, the postoperative ventilation cutoff (86.5 min) seems unusually brief clinically; clarify if units were misreported (minutes vs hours). Additionally, multicollinearity between biomarkers such as WBC/PLT and Ne/PLT is possible; include correlation analysis or justify their simultaneous use. Lastly, discuss potential biases due to retrospective single-center design and validate externally if feasible. Recommendation: revise.

(1)Please explain why $7:3$ is used as the training sample to validation sample ratio. Will such a ratio cause over-fitting problem?

Response #2: We thank the reviewer for raising this critical methodological consideration. The rationale for adopting a 7:3 training-validation split in our study is twofold. Firstly, with a limited sample size (n<1000), a 7:3 ratio ensures sufficient training data for robust parameter estimation while retaining a meaningful subset for preliminary validation. This approach balances the dual objectives of model learning and performance assessment under resource constraints. Secondly, consistent with prior studies[1,2], we employed a 7:3 ratio, adhering to this empirical convention. While split ratios influence validation reliability, overfitting is predominantly governed by: sample size adequacy, model complexity and data quality. Thank you for your comments.

(2)Firstly, the selection criteria for biomarkers appear arbitrary (AUC > 0.6), lacking robust justification. Clearly define the rationale behind this threshold or consider adjusting based on clinical relevance.

Response #2: Thank you for the reviewer's comments, which are helpful for the improvement of our research. As can be seen from the article, this is a process of gradually screening variables. In the first round of screening, we used ROC to screen all biomarkers with the aim of selecting variables that are relatively related to the outcome, as well as choosing the best cutoff value, thus selecting the value of 0.6. This is because it is not very strict but still has distinguishing ability. Thank you for your comments.

(3)Secondly, the postoperative ventilation cutoff (86.5 min) seems unusually brief clinically; clarify if units were misreported (minutes vs hours).

Response #2: We sincerely appreciate the reviewer's insightful observation regarding the mechanical ventilation duration. The original unit was indeed reported incorrectly. We have made the following corrections: The postoperative ventilation cutoff value has been revised from 86.5 minutes to 86.5 hours throughout the manuscript (including Abstract, Results, Discussion, Conclusion, Table 2 and Table 3). We reconfirmed the raw data from anesthesia records and ICU flow sheets to ensure accuracy. The revised cutoff now appropriately stratifies patients into risk groups for subsequent analyses. These changes have been highlighted in

---

## [Decision Letter · Decision Letter 1]

16 Jun 2025

Dear Dr. chen,

Thank you for submitting your manuscript to PLOS ONE. After careful consideration, we feel that it has merit but does not fully meet PLOS ONE’s publication criteria as it currently stands. Therefore, we invite you to submit a revised version of the manuscript that addresses the points raised during the review process.

We look forward to receiving your revised manuscript.

Kind regards,

Tomasz W. Kaminski

Academic Editor

PLOS ONE

Additional Editor Comments:

Dear Authors,

Thank you for submitting the revised version of your manuscript. We appreciate the effort you put into addressing the reviewers’ comments.

While your revisions have improved the manuscript, one of the reviewers remains concerned about several key issues that they feel were not adequately resolved. Although the second reviewer declined to re-review, your responses to their comments appear reasonable and have full merit.

Given the nature of the remaining concerns, we believe that an additional round of revision is necessary before the manuscript can proceed further in the review process. We encourage you to carefully address all of Reviewer 2’s points in detail in your next revision.

Thank you for your continued engagement, and we look forward to receiving your revised manuscript.

Best regards.

Reviewers' comments:

Reviewer's Responses to Questions

**Comments to the Author**

Reviewer #2: All comments have been addressed

2. Is the manuscript technically sound, and do the data support the conclusions?

Reviewer #2: Partly

3. Has the statistical analysis been performed appropriately and rigorously?

Reviewer #2: Yes

4. Have the authors made all data underlying the findings in their manuscript fully available?

Reviewer #2: Yes

5. Is the manuscript presented in an intelligible fashion and written in standard English?

Reviewer #2: Yes

Reviewer #2: The revised manuscript shows some improvements (e.g., inclusion of multicollinearity diagnostics and correction of ventilator-time units) but key methodological issues remain. This includes the arbitrary choice of AUC > 0.6 for biomarker selection and dichotomization of continuous predictors, which is poorly justified; such cutoffs discard information and can bias results

pubmed.ncbi.nlm.nih.gov

Citing "clinical convenience" alone is not convincing, as the clinical validity of the chosen thresholds is unclear. Potential interactions among predictors were not examined. The mixed use of significance thresholds (α = 0.05 vs. 0.1) is unconventional and should be clarified. Similarly, the fixed 7:3 train–test split lacks strong rationale; cross-validation or bootstrapping might better exploit the limited data. External validation is still absent, leaving generalizability untested. For example, performance metrics (e.g., AUC and calibration) are not reported, so predictive accuracy remains unclear. In its current form, I recommend major revision.

**Do you want your identity to be public for this peer review?** For information about this choice, including consent withdrawal, please see our Privacy Policy

Reviewer #2: **Yes: ** Robert J. Chen, MD, MPH

---

## [Author Response · Author response to Decision Letter 2]

27 Jun 2025

Dear Editors and Reviewers: 

Thank you for your letter and the reviewers' comments on our manuscript "Prognostic nomogram integrated with inflammatory marker ratios for assessing in-hospital mortality risk in patients with acute type A aortic dissection" (PONE-D-25-13134). We sincerely thank the reviewer for their comments in this second round. We provide the following responses to the reviewer's points and respectfully request the editor's careful consideration of our explanations.

Reviewers' comments:

1. If the authors have adequately addressed your comments raised in a previous round of review and you feel that this manuscript is now acceptable for publication, you may indicate that here to bypass the “Comments to the Author” section, enter your conflict of interest statement in the “Confidential to Editor” section, and submit your "Accept" recommendation.

Reviewer #2: All comments have been addressed.

Response #2: Thanks.

2. Is the manuscript technically sound, and do the data support the conclusions?

Reviewer #2: Partly

Response #2: Thank you for the reviewers' comments.

3.Has the statistical analysis been performed appropriately and rigorously?

Reviewer #2: Yes

Response #2: Thanks.

4.Have the authors made all data underlying the findings in their manuscript fully available?

Reviewer #2: Yes

Response #2: Thanks.

5. Is the manuscript presented in an intelligible fashion and written in standard English?

Reviewer #2: Yes

Response #2: Thanks.

6. Review Comments to the Author

Reviewer #2: The revised manuscript shows some improvements (e.g., inclusion of multicollinearity diagnostics and correction of ventilator-time units) but key methodological issues remain. This includes the arbitrary choice of AUC > 0.6 for biomarker selection and dichotomization of continuous predictors, which is poorly justified; such cutoffs discard information and can bias results pubmed.ncbi.nlm.nih.gov. Citing "clinical convenience" alone is not convincing, as the clinical validity of the chosen thresholds is unclear. Potential interactions among predictors were not examined. The mixed use of significance thresholds (α = 0.05 vs. 0.1) is unconventional and should be clarified. Similarly, the fixed 7:3 train–test split lacks strong rationale; cross-validation or bootstrapping might better exploit the limited data. External validation is still absent, leaving generalizability untested. For example, performance metrics (e.g., AUC and calibration) are not reported, so predictive accuracy remains unclear. In its current form, I recommend major revision.

Response #2: Thank you for the reviewers' comments. The reviewers provided professional opinions on statistical analysis, which were helpful for the improvement of our research.

(1)Regarding the suggestion to use cross-validation for internal validation to utilize existing data more effectively: This is an excellent suggestion. We have supplemented the manuscript accordingly. We employed 10-fold cross-validation to further evaluate the model. The results demonstrate that the model possesses good discriminative performance, with a calibrated C-index of 0.807.

(2)Regarding the concern about converting continuous biomarker values into binary categorical variables: However, this approach is common practice in many studies. For example, in predictive models for hepatocellular carcinoma (HCC) prognosis, cutoff values for AFP (Alpha-fetoprotein) such as 200, 400, or even 800 ng/mL are frequently used. Numerous studies utilize the cutoff method to transform continuous variables into binary categories when establishing thresholds.

1)Concerning AUC threshold selection: Both thresholds of 0.6 and 0.7 are used in existing research. We opted for 0.6 to ensure a broader range of biomarkers entered the univariate analysis, preventing potentially useful clinical indicators from being overlooked.

2)Clinical validity of the threshold: The AUC value demonstrates the clinical validity of the chosen threshold. An AUC > 0.6 indicates the model's predictive ability is superior to standard random guessing (AUC = 0.5), meaning the variable can differentiate between the two classes to a certain extent. Whether it progresses to the final predictive model depends on the results of subsequent analyses.

3)Interaction between predictors: Potential interactions were also assessed through collinearity diagnostics.

(3)Regarding the choice of variable selection method and data splitting: We selected a classic stepwise selection method, which is widely applied in clinical prediction models. The use of mixed significance thresholds and a fixed 7:3 training-test set split are commonly employed in numerous studies. Therefore, we believe these aspects can be retained without modification (Supporting References: DOI: 10.1007/s00431-022-04761-9; DOI: 10.1186/s40001-023-01255-8; DOI: 10.3389/fimmu.2024.1437980). Please give our viewpoints full consideration. Once again, thank you very much for your comments and suggestions.

---

## [Decision Letter · Decision Letter 2]

9 Jul 2025

Dear Dr. chen,

Thank you for submitting your manuscript to PLOS ONE. After careful consideration, we feel that it has merit but does not fully meet PLOS ONE’s publication criteria as it currently stands. Therefore, we invite you to submit a revised version of the manuscript that addresses the points raised during the review process.

We look forward to receiving your revised manuscript.

Kind regards,

Tomasz W. Kaminski

Academic Editor

PLOS ONE

Additional Editor Comments:

Dear Authors,

Thank you for your revision. Unfortunately, the main issues previously raised remain unaddressed.

Both the original and an additional reviewer agree that the manuscript is undermined by critical methodological flaws - most notably, the use of data-driven dichotomization of continuous variables via ROC analysis. This practice is statistically invalid, discards information, and introduces bias. The defense of this approach as “common practice” reflects a misunderstanding of core biostatistical principles.

Further concerns include misinterpretation of collinearity checks as interaction testing, lack of internal or external validation, non-compliance with TRIPOD standards, and the use of causal language unsupported by the study design.

A full re-analysis is necessary: continuous modeling of predictors, formal interaction testing, internal validation (e.g., bootstrapping), and adherence to TRIPOD guidelines - as mentioned by the Reviewer 2. Without these changes, the paper cannot be accepted.

Reviewers' comments:

Reviewer's Responses to Questions

**Comments to the Author**

Reviewer #2: (No Response)

2. Is the manuscript technically sound, and do the data support the conclusions?

Reviewer #2: Partly

3. Has the statistical analysis been performed appropriately and rigorously?

Reviewer #2: No

4. Have the authors made all data underlying the findings in their manuscript fully available?

Reviewer #2: Yes

5. Is the manuscript presented in an intelligible fashion and written in standard English?

Reviewer #2: No

Reviewer #2: This manuscript, developing a prognostic nomogram for aortic dissection, is invalidated by fundamental methodological errors. Its primary flaw is dichotomizing continuous predictors using data-dredged "optimal" cut-points from ROC analysis—a statistically invalid practice that discards information and creates bias. The author's reply, defending this as "common practice," reveals a critical misunderstanding of biostatistical principles. The study also incorrectly equates collinearity diagnostics with testing for interactions, lacks external validation, and fails to meet TRIPOD reporting standards. The English is imprecise, using causal language unsupported by the design. A complete re-analysis is mandatory: predictors must be modeled continuously, interactions formally tested, and the model validated using bootstrapping, with full adherence to TRIPOD guidelines.

**Do you want your identity to be public for this peer review?** For information about this choice, including consent withdrawal, please see our Privacy Policy

Reviewer #2: **Yes: ** Robert J. Chen, MD, MPH

---

## [Author Response · Author response to Decision Letter 3]

7 Aug 2025

Dear Editors and Reviewers: 

Thank you for your letter and the reviewers' comments on our manuscript "Prognostic nomogram integrated with inflammatory marker ratios for assessing in-hospital mortality risk in patients with acute type A aortic dissection" (PONE-D-25-13134). We sincerely thank the reviewer for their comments in this third round. We have studied comments carefully and the responds to the reviewer’s comments are as following:

Additional Editor Comments: Both the original and an additional reviewer agree that the manuscript is undermined by critical methodological flaws - most notably, the use of data-driven dichotomization of continuous variables via ROC analysis. This practice is statistically invalid, discards information, and introduces bias. The defense of this approach as “common practice” reflects a misunderstanding of core biostatistical principles.

Further concerns include misinterpretation of collinearity checks as interaction testing, lack of internal or external validation, non-compliance with TRIPOD standards, and the use of causal language unsupported by the study design.

A full re-analysis is necessary: continuous modeling of predictors, formal interaction testing, internal validation (e.g., bootstrapping), and adherence to TRIPOD guidelines - as mentioned by the Reviewer 2. Without these changes, the paper cannot be accepted.

Response: We sincerely thank the reviewers and editors for their rigorous critique. We fully acknowledge the critical methodological flaws identified, particularly the inappropriate dichotomization of continuous variables and insufficient validation. We unequivocally accept all criticisms and have implemented comprehensive revisions, such as retaining continuous variables in their original form without dichotomization, with detailed measures specified in our point-by-point response to Reviewer 2.

Reviewers' comments:

Comments to the Author

1. Is the manuscript technically sound, and do the data support the conclusions?

Reviewer #2: Partly

Response:Thank you for the reviewers' comments.

2. Has the statistical analysis been performed appropriately and rigorously?

Reviewer #2: No

Response:Thank you for your valuable comments on the rigor of statistical analysis. We have carefully reviewed the Methods section, and made point-by-point revisions in response to the reviewers' comments.

3. Have the authors made all data underlying the findings in their manuscript fully available?

Reviewer #2: Yes

Response:Thanks.

4. Is the manuscript presented in an intelligible fashion and written in standard English?

Reviewer #2: No

Response: We thank the reviewer for highlighting the language concerns. We have revised the language throughout the manuscript, including correcting grammatical and syntactical errors, to meet PLOS ONE's publication standards.

5. Review Comments to the Author

Reviewer #2: This manuscript, developing a prognostic nomogram for aortic dissection, is invalidated by fundamental methodological errors. Its primary flaw is dichotomizing continuous predictors using data-dredged "optimal" cut-points from ROC analysis—a statistically invalid practice that discards information and creates bias. The author's reply, defending this as "common practice," reveals a critical misunderstanding of biostatistical principles. The study also incorrectly equates collinearity diagnostics with testing for interactions, lacks external validation, and fails to meet TRIPOD reporting standards. The English is imprecise, using causal language unsupported by the design. A complete re-analysis is mandatory: predictors must be modeled continuously, interactions formally tested, and the model validated using bootstrapping, with full adherence to TRIPOD guidelines.

Response #2: We sincerely thank Reviewer #2 for their rigorous and constructive critique, which has highlighted critical methodological shortcomings in our study. We fully acknowledge the validity of the concerns raised. Based on the reviewers' comments and our team's discussion, we have comprehensively revised our manuscript by selecting clinically relevant inflammatory markers including white blood cell count, neutrophil count, hemoglobin, D-dimer, blood urea nitrogen, albumin, and neutrophil-to-lymphocyte ratio; to prevent loss of information, we preserved continuous variables in their original form rather than dichotomizing them (e.g., treating values >50 as equivalent when 51 and 99 represent distinct clinical states); additionally, to avoid multicollinearity, we excluded blood cell parameters used to derive inflammatory indices and calculated variance inflation factors (VIF < 5) for included variables; finally, due to unavailable external validation cohorts, we implemented internal validation through bootstrapping while explicitly acknowledging this limitation in the Discussion. We extend sincere appreciation to the handling editor and reviewers for their guidance.

Once again, thank you very much for your comments and suggestions.

---

## [Decision Letter · Decision Letter 3]

27 Aug 2025

Dear Dr. chen,

Thank you for submitting your manuscript to PLOS ONE. After careful consideration, we feel that it has merit but does not fully meet PLOS ONE’s publication criteria as it currently stands. Therefore, we invite you to submit a revised version of the manuscript that addresses the points raised during the review process.

We look forward to receiving your revised manuscript.

Kind regards,

Tomasz W. Kaminski

Academic Editor

PLOS ONE

Journal Requirements:

Additional Editor Comments:

Dear Authors,

Thank you for your careful and honest work in revising the manuscript. You have addressed the major methodological concerns. The revised model is methodologically sound and clearly presented.

What remains are only minor points - softening causal phrasing, emphasizing the lack of external validation, and some residual English editing. With these small adjustments, your manuscript will be ready for acceptance.

Best regards,

Tomasz W Kaminski

Reviewers' comments:

Reviewer's Responses to Questions

**Comments to the Author**

Reviewer #2: (No Response)

2. Is the manuscript technically sound, and do the data support the conclusions?

Reviewer #2: Yes

3. Has the statistical analysis been performed appropriately and rigorously?

Reviewer #2: Yes

4. Have the authors made all data underlying the findings in their manuscript fully available?

Reviewer #2: Yes

5. Is the manuscript presented in an intelligible fashion and written in standard English?

Reviewer #2: No

Reviewer #2: The authors have implemented the key corrections requested in the earlier rounds. The major concern—ROC-based dichotomization—has now been addressed, and variables are retained in their continuous form. Multicollinearity was formally checked with VIF, and internal validation by bootstrapping was performed, with limitations clearly acknowledged. The revised modeling strategy (Hb, NLR, and MODS) is methodologically sound and presented transparently.

Remaining issues are minor. The paper still uses slightly casual causal phrasing in places (“guiding therapy,” “improving prognosis”), which should be softened to “may assist” or “is associated with.” Also, while external validation is understandably not possible, this limitation should be emphasized again in the discussion or acknowledgments. Finally, English clarity has improved, but residual phrasing errors persist (e.g., “conduct more accurate multi-indicator joint individualized predictions”) that could benefit from professional editing.

Overall, the fatal flaws have been corrected. With minor language polishing and acknowledgment of external validation limits, the manuscript is acceptable.

**Do you want your identity to be public for this peer review?** For information about this choice, including consent withdrawal, please see our Privacy Policy

Reviewer #2: **Yes: ** Robert J. Chen, MD, MPH

---

## [Author Response · Author response to Decision Letter 4]

5 Sep 2025

Dear Editors and Reviewers: 

Thank you for giving us the opportunity to revise our manuscript [PONE-D-25-13134] entitled “Prognostic nomogram integrated with inflammatory marker ratios for assessing in-hospital mortality risk in patients with acute type A aortic dissection”. We are grateful to the editor and the reviewers for their time and insightful comments, which have helped us to improve the quality of our paper. We have carefully addressed all the points raised in the review.

Journal Requirements:

Response: Thank you for the suggestion. We have conducted a thorough review of the reference list to further enhance the academic rigor of our manuscript. During this process, we noted that the reference "[24] Zhang Y, Chen T, Chen Q, Min H, Nan J, Guo Z. Development and evaluation of an early death risk 364 prediction model after acute type A aortic dissection. Ann Transl Med. 2021 Sep;9(18):1442. doi: 365 10.21037/atm-21-4063." (originally cited at [Line 364-366]), while relevant, was not indexed in SCI and its supporting evidence was somewhat limited. To more robustly substantiate our argument and align with the highest scholarly standards, we have replaced it with a more authoritative publication from a SCI-indexed journal: "[24]Moisi MI, Bungau SG, Vesa CM, Diaconu CC, Behl T, Stoicescu M, et al. Framing Cause-Effect Relationship of Acute Coronary Syndrome in Patients with Chronic Kidney Disease. Diagnostics (Basel). 2021 Aug 23;11(8):1518. doi: 10.3390/diagnostics11081518.". This newer reference provides stronger and more compelling evidence for the point discussed, and the change has been incorporated at [Line 365-367] in the revised manuscript.

Furthermore, we have performed a comprehensive check of the entire reference list to ensure its full compliance with the journal's guidelines. This includes verifying the accuracy and completeness of all citations, confirming that no retracted papers are cited, and ensuring that the references are up-to-date and encompass the most relevant literature in the field.

Review Comments to the Author

Reviewer #2: The authors have implemented the key corrections requested in the earlier rounds. The major concern-ROC-based dichotomization-has now been addressed, and variables are retained in their continuous form. Multicollinearity was formally checked with VIF, and internal validation by bootstrapping was performed, with limitations clearly acknowledged. The revised modeling strategy (Hb, NLR, and MODS) is methodologically sound and presented transparently.

Remaining issues are minor. The paper still uses slightly casual causal phrasing in places (“guiding therapy,” “improving prognosis”), which should be softened to “may assist” or “is associated with.” Also, while external validation is understandably not possible, this limitation should be emphasized again in the discussion or acknowledgments. Finally, English clarity has improved, but residual phrasing errors persist (e.g., “conduct more accurate multi-indicator joint individualized predictions”) that could benefit from professional editing.

Overall, the fatal flaws have been corrected. With minor language polishing and acknowledgment of external validation limits, the manuscript is acceptable.

Response: We sincerely thank Reviewer #2 for their positive feedback and acknowledgment that our revisions have addressed the major methodological concerns. We are also grateful for the constructive suggestions regarding language and clarity. We have carefully implemented all remaining recommendations as follows:

(1)Causal phrasing: All instances of overly strong language (e.g., "guiding therapy," "improving prognosis") have been softened throughout the manuscript to more cautious and accurate terms.

(2)Emphasis on external validation limitation: As suggested, we have further emphasized the absence of external validation as a key limitation in the Limitation section.

(3)Professional English editing: The manuscript has been professionally edited for English language by American Journal Experts (Certificate Number: 98BA-E44B-9F71-2820-0439). All suggested changes have been reviewed and incorporated by the authors to ensure clarity and readability.

We thank the reviewer again for their thorough and helpful assessment of our work.

---

## [Editor Report · Decision Letter 4]

9 Sep 2025

Prognostic nomogram integrated with inflammatory marker ratios for assessing in-hospital mortality risk in patients with acute type A aortic dissection

PONE-D-25-13134R4

Dear Dr. chen,

We’re pleased to inform you that your manuscript has been judged scientifically suitable for publication and will be formally accepted for publication once it meets all outstanding technical requirements.

Kind regards,

Tomasz W. Kaminski

Academic Editor

PLOS ONE

---

## [Editor Report · Acceptance letter]

PONE-D-25-13134R4

PLOS ONE

Dear Dr. Chen,

I'm pleased to inform you that your manuscript has been deemed suitable for publication in PLOS ONE. Congratulations! Your manuscript is now being handed over to our production team.

Kind regards,

on behalf of

Dr. Tomasz W. Kaminski

Academic Editor

PLOS ONE